# A comparative study on fatty acid profile in selected vessels of coronary artery bypass graft (CABG)

E. M. S. Bandara[1], D. I. U. Edirisinghe[2], D. D. C. de S. Wanniarachchi[2]*, H. Peiris[3], P. P. R. Perera[3], A. G. Jayakrishan[4], H. D. Waikar[4], S. K. Sharma[4], V. Abeysuriya[4], L. G. Chandrasena[4]

1 Department of Medical Laboratory Science, Faculty of Allied Health Sciences, University of Sri Jayewardenepura, Nugegoda, Sri Lanka, 2 Instrument Center, Faculty of Applied Sciences, University of Sri Jayewardenepura, Nugegoda, Sri Lanka, 3 Department of Biochemistry, Faculty of Medical Sciences, University of Sri Jayewardenepura, Nugegoda, Sri Lanka, 4 Nawaloka Hospital Research and Education Foundation, Nawaloka Hospitals PLC, Colombo, Sri Lanka

* dakshikacw@sjp.ac.lk

**Data Availability Statement:** All relevant data are within the manuscript and its Supporting Information files.

## Abstract

Atherosclerosis is one of the leading non-communicable diseases in Sri Lanka. Analysis of fatty acid composition in blood vessels is important in understanding the development of atherosclerosis. Here, analyses of fatty acid profiles in major arteries which are commonly used in Coronary Artery Bypass Graft surgery (CABG) were subjected to investigation. Patients (n = 27) undergoing elective CABG were enrolled in the study. A small biopsy segment of the saphenous vein (SV), radial artery (RA), and left internal mammary artery (LIMA) of patients was obtained during the surgery. The fatty acid (FA) profile of tissue samples was analyzed using Gas Chromatography-Mass Spectroscopy (GCMS). Among the different arteries tested, palmitic acid and stearic acid were the predominant fatty acids. As far as monounsaturated FA (MUFA) are concerned, oleic acid was found to be the most abundant MUFA in vessels. The FA profile of LIMA samples had a higher SFA percentage and lower unsaturated FA percentage compared to other vessels. Furthermore, the vessel samples of RA indicated the highest percentage of pro-inflammatory ω -6 polyunsaturated fatty acids (PUFA) as well as a higher percentage ratio between ω -6: ω -3 PUFA. The fatty acid composition and ω -6: ω -3 PUFA ratio suggests that LIMA graft is preferred for CABG over RA and SV.

## Introduction

Ischemic heart disease is the leading cause of death worldwide [1]. The success of coronary artery bypass grafting (CABG) which is one of the common surgical techniques used to improve the blood flow in the heart is dependent on the long-term patency of arterial and venous grafts [2]. The majority of patients receive left internal mammary artery (LIMA) grafts to the left anterior descending (LAD) coronary artery compared to other grafting options such

**Funding:** The work was supported by an institutional research grant from the University of Sri Jayewardenepura (Grant number ASP/01/RE/SCI/2019/67) and funding received from Nawaloka Hospital Research and Education Foundation, Nawaloka Hospitals PLC, Colombo, Sri Lanka. The funders had no role in study design, data collection and analysis, decision to publish, or preparation of the manuscript.

**Competing interests:** The authors have declared that no competing interests exist.

**Abbreviations:** CABG, Coronary artery bypass graft; SV, Saphenous vein; RA, Radial artery; LIMA, Left internal mammary artery; GCMS, Gas chromatography mass spectroscopy; MUFA, Monounsaturated fatty acid; PUFA, Polyunsaturated fatty acid.

as saphenous vein (SV) grafts or radial artery (RA) [3]. According to literature, the LIMA is considered a superior graft for long-term patency when compared to a vein graft [4]. Both histological and chemical composition of grafts could have influences on early proatherogenic changes and late graft failure 3–5 years of surgery [5, 6].

Fatty acid composition of vessel walls is an important modulator in vascular function [7]. Changes in fatty acid composition can modify the properties of cell membranes including fluidity and permeability, either directly via their insertion in the membrane or indirectly through their effects on lipid metabolism [8]. Hence, several studies have observed the influence of the vessel wall fatty acid composition in atherosclerosis [9, 10].

It has been found that inflammation present in advanced plaques can lead to critical cardiovascular events, thus fatty acid composition and inflammatory cells in vessel walls are key to plaque stability and morphology [11]. Furthermore, contribution to inflammation from eicosanoids originating from arachidonic acid can be controlled by increasing the levels of EPA and DHA in cells of vessel walls [11]. Therefore, the ratio of such ω-6 fatty acids to ω-3 fatty acids is important in the determination of inflammation [12]. Though there are abundant studies correlating atherosclerosis with plasma fatty acid composition [13, 14], only limited studies have been conducted on the analysis of the vessel wall fatty acid compositions. A comparative study conducted on aorta and LIMA wall fatty acid profiles have found differences in the fatty acid profile of aorta and LIMA and suggest individual fatty acids as a contributing factor for atherogenesis by observing lower levels of unsaturated fatty acids and higher amounts of saturated fatty acids in the atherosclerotic aorta [10]. Similarly, high levels of polyunsaturated fatty acids have been observed in SV compared to aortic tissue [15].

These findings emphasize the importance of fatty acid profiles in atherosclerotic tissues and their role in atherogenesis and coronary artery disease (CAD). Even though the fatty acid profile of SV, and LIMA had been studied before [10, 15], there was no information on the comparison of these profiles with radial arteries. Therefore, this study aims to compare the fatty acid profile of common vessels used in CABG. Such comparison of fatty acids present in different vessels will aid in the understanding the association between fatty acid profile and atherosclerotic plaque redevelopment.

## Materials and methods

### Selection of CAD patients

Twenty-seven angiographically proven coronary artery disease patients were enrolled in this study. The sample size was calculated using a single group standard deviation of Docosahexaenoic acid level of the internal mammary artery [10]. The subjects were accepted for elective CABG at Nawaloka Hospital, Colombo, Sri Lanka. Biopsy samples (1–2) g of LIMA (12 samples), SV (20 samples), and RA (9 samples) were obtained during surgery (Total 41 samples). Ethical clearance (Ref. 34/19) was obtained from the ethics review committee of the Faculty of Medical Sciences, University of Sri Jayewardenepura. The study was conducted according to the guidelines set by the Declaration of Helsinki. Informed written consent from all participants was obtained prior to recruitment to the study.

### Fatty acid analysis

After sampling the specimens were preserved in methanol and kept in a refrigerator (2–8°C) until analysis. The fatty acid extraction was mainly based on the Folch method [16]. Each tissue sample was homogenized with chloroform (SRL, reagent grade): methanol (SRL, reagent grade) mixture (2:1 v/v) to a final 20-fold dilution and methanol used in preservation was used here. The sample was washed with 0.9% NaCl solution and was followed by centrifugation.

The upper phase was discarded and the lower layer which contains the fat extract was evaporated to near dryness. The extracted fatty acids were methylated in the presence of the base catalyst KOH (Sigma Aldrich) in methanol. The Fatty acid methyl esters were then extracted to hexane (SRL, HPLC grade) by centrifuging. Gas Chromatography-Mass Spectroscopy (GCMS) analysis was performed on Agilent 7890A GC coupled to 5975C MS with Triple-Axis Detector using a split /splitless inlet with He as the carrier gas with 1 mL/min rate. The HP 5MS column with dimensions 30m, 0.25mm, 0.25um was used for analysis. The oven temperature program was as follows: the column was initially held at 90 ˚C for 1 min after injection, then increased to 150 ˚C with a 12 ˚C/min heating ramp for 5 min. Then increased to 180 ˚C with a 5 ˚C/min heating ramp for 6 min and increased to 210 ˚C with a 3 ˚C/min heating ramp for 10 min and finally to a temperature of 250 ˚C with a rate of 10 ˚C/min for 4 min. The total run time was 39 min. The injector port temperature was held at 270 ˚C. MS source temperature was 230 ˚C and MS Quadrupole temperature was at 150 ˚C and mass range analysis was at 50–550 m/z. The compound identification was done comparing MS results with NIST database Chemstation® software. The percentage of individual fatty acids was calculated as the area percentage of each fatty acid to the total area of all fatty acids detected.

## Statistical analysis

The data was analyzed using SPSS version 25. Since the Kolmogorov–Smirnov test reported that the variables are not normally distributed, Mann–Whitney U-tests were used to compare the fatty acid percentage of LIMA, SV, and RA. A p-value of $< 0.05$ was considered to be significant. A linear regression model was developed to determine the association of confounding variables with fatty acid levels. The results were reported as mean ± standard deviation (SD).

## Results

The demographic and clinical characteristics of the study subjects are shown in Table 1. The study included 27 patients with an average age of 58 years and 85% of the study population were male. Almost half of the patients recruited were diabetic (53%), 74.1% were hypertensive and 84.1% were diagnosed as hyperlipidemic. The average lipid profile parameters of the study group were within normal reference ranges. More than 2/3 of individuals had myocardial infarction prior to CABG. Out of 23 males, 20 individuals had history of smoking and 05 of them are still continuing occasionally. The estimated 10-year primary risk of atherosclerotic cardiovascular disease (ASCVD) among patients was estimated and was found to be $> 7.5\%$ for all. Out of sample, 20 individuals were in high-risk group and 07 in the intermediate-risk group [17].

The percentage of individual fatty acids is given in Table 2 and fatty acid profiles of the three samples LIMA, SV, and RA have been compared. The percentages of fatty acids are given in S1 Table for LIMA, S2 Table for SV, and S3 Table for RA.

The comparison of the fatty acid profile of LIMA with other vessels indicates a significantly high percentage of lauric acid (C12:0) compared to SV (p = 0.001) and RA (p = 0.036). A significantly lower arachidonic acid (AA) percentage was observed in LIMA compared to SV (p = 0.018) and RA (p = 0.019). The only ω-3, PUFA reported was Docosahexaenoic acid (DHA), with the lowest percentage being found in RA.

In order to obtain a summary of the above variations, the levels of saturated (SFA) and unsaturated fatty acids(USFA) were analyzed and the results are given in Table 3. In LIMA samples, SFA content was higher than the USFA. In contrast, SV and RA contained a higher proportion of USFA than SFA. The Ratio of AA, ω-6/ DHA, ω-3 was significantly higher in RA compared to SV and LIMA.

**Table 1. Demographic and clinical characteristics of patients in the study.**

| CAD patients (n = 27) | |
|---|---|
| Age (y) | 57.9 ± 8.3 |
| Gender (n) | |
| Male | 23 |
| Female | 4 |
| Hypertension (%) | 74.1% |
| Diabetes (%) | 55.5% |
| Hyperlipidemia | 81.4% |
| Total Cholesterol (mg/dL) | 162 ± 36 |
| LDL | 104 ± 40 |
| HDL | 36 ± 8 |
| Triglycerides | 145 ± 40 |
| Statin use Atorvastatin 40 mg daily dose | 92.6% |
| Atorvastatin 40 mg daily + Ezetimibe 10 mg | 7.4% |
| Family history of CAD | 59.3% |
| Prior Myocardial Infarction | 70.3% |
| History of Smoking (n = 20/27) | 74.0% |
| Previously smoked (n = 15/20) | 75% |
| Currently smoking (n = 5/20) | 25% |
| ASCVD risk estimation (range %) | |
| Low (< 5%) | - |
| Borderline (5–7.4%) | - |
| Intermediate (7.5–19.9%) n = 7 | 8.5–18.2% |
| High risk (> 20%) n = 20 | 25–47% |

The linear regression models were developed for fatty acid with significant variance to determine the effect of confounding factors and the results are as given in S4 Table. The factors considered were age, gender, and presence of diabetes mellitus, hypertension, hyperlipidemia, family history of CAD, smoking, and lipid profile parameters. The results revealed that family history of CAD was significantly associated with a lauric acid level of SV (B = 1.922, p = 0.05,

**Table 2. Fatty acid composition (Mean%±SD) of LIMA, SV, and radial artery in patients (n = 27) with CAD.**

| Fatty acid | Segment | | |
|---|---|---|---|
| | LIMA | SV | RA |
| 12:0 (Lauric acid) | 6.49 ±3.40* | 3.41 ± 4.12 | 3.32 ± 2.43 |
| 14:0 (Myristic acid) | 9.18 ±4.67 | 7.15 ± 10.14 | 7.33 ± 2.79 |
| 16:0 (Palmitic acid) | 36.04 ± 13.57 | 30.44± 11.42 | 31.84 ±12.20 |
| 16:1 (Palmitoleic acid) | 6.66 ± 5.76 | 7.61 ± 4.37 | 5.39 ±2.43 |
| 18:0 (Stearic acid) | 6.45 ± 4.95 | 9.62 ± 11.10 | 5.78 ±2.19 |
| 18:1 (Oleic acid) | 31.76 ± 13.96 | 40.30± 20.61 | 40.59 ±8.25 |
| 18:2, ω-6 (Linoleic acid) | 5.96 ± 4.08 | 3.78 ± 2.44 | 5.79 ±2.79 |
| 20:4, ω-6 (Arachidonic acid) | 0.49 ± 0.42* | 4.44 ± 3.69 | 1.72 ±0.92 |
| 22:6, ω-3 (Docosahexaenoic acid) | 0.41 ± 0.29 | 1.66 ±1.24 | 0.27 ±0.06 |

* $p < 0.05$, mean values are significantly different compared to SV and RA

Palmitic acid (16:0) was the major fatty acid in all the tissues followed by oleic acid (18:1) and stearic acid (18:0). The percentage concentration of the monounsaturated fatty acid, oleic acid (18:1) was comparatively higher in SV and RA than LIMA.

**Table 3. Summary of main categories of fatty acids in LIMA, SV and RA.**

| Fatty acid | Segment | | |
|---|---|---|---|
| | LIMA (Mean%±SD) | SV (Mean%±SD) | RA (Mean%±SD) |
| Saturated FA (SFA) | 54.63± 12.8 | 45.65±16.06 | 48.27± 9.5 |
| Unsaturated FA (MUFA + PUFA) | 45.37± 12.8 | 54.35± 16.0 | 51.73± 9.5 |
| USFA: SFA | 0.95± 0.65 | 1.8 ± 2.4 | 1.16 ± 0.53 |
| PUFA, ω-6 | 5.2 ± 2.3 | 5.7 ± 4.2 | 6.3 ± 3.5 |
| AA, ω-6/ DHA, ω-3 | 1.8 ± 0.57 | 2.9 ± 0.9 | 5.4 ± 0.6* |

*$p < 0.05$, mean value is significantly different compared to SV and LIMA.

95% CI; -0.12–3.85). Family history of CAD (B = 2.996, p = 0.026, 95% CI; 0.447–5.54) and smoking (B = 3.642, p = 0.039, 95% CI; 0.225–7.06) were significant factors in associated with arachidonic acid level of saphenous vein. Further, gender was a significant (B = 0.409, p = 0.044, 95% CI; 0.013–0.804) confounding factor for increased arachidonic acid level in LIMA.

## Discussion

The present study reports the comparison of the fatty acid composition of three biopsy specimens of three important vessels collected during CABG, namely LIMA, RA, and SV for the first time in literature. The FA profile of vessels that are implanted during CABG can have an effect on long-term stability. The results were compared with similar studies found in the literature where two vessels have been compared.

Studies conducted by Oskouei and coworkers have compared the fatty acid composition of aorta and SV and indicated that SV had a higher percentage of PUFA [15]. The percentages of major FA in aortic samples indicated higher amounts of palmitic acid and lower amounts of oleic acid compared to SV. The fatty acid profiles of LIMA and aorta were compared by Bahrami *et al* and the results indicate higher SFA content in the aorta than the LIMA with palmitic acid and stearic acid as the major SFA [10]. Therefore, FA profiles of SV and LIMA indicate lower SFA content than aorta in literature. According to observations in this study, SV contained a high percentage of USFA and a low percentage of SFA compared to both RA and LIMA. Although the SFA content was high for RA than the SV yet it was lower than in LIMA. Since palmitic acid percentages are almost the same, higher stearic acid content observed in RA results in a higher percentage of SFA. The opposite trend was observed for USFA in RA.

When considering the long-term success of the grafted vessels, it is important to prevent the redevelopment of atherosclerotic plaque [18]. Plaque develops as a consequence of endothelial injury and activation, monocyte accumulation and formation of foam cells, lipid accumulation and smooth muscle proliferation (atherogenesis), extreme inflammation. Plaque progression and plaque rupture ultimately cause thrombogenesis and obstruct the blood flow to cardiac muscles [19]. Therefore, it is worth considering the role of ω-3 and ω-6 fatty acid levels in the redevelopment of atherosclerotic plaque [20].

Several studies have reported the effects of ω−3 fatty acids on the improvement of vascular structure and function. Reduced LDL uptake in the arterial wall is a reported function of ω−3 fatty acids which is achieved by reducing expression of lipoprotein lipase [21]. The effect of ω−3 fatty acids on smooth muscle cells has been studied in cell lines and human coronary arteries. Smooth muscle cell proliferation has been suppressed through inhibition of DNA synthesis and replication in ω−3 fatty acid (DHA and EPA) incorporated cell lines [22]. Similar observations are also reported in human coronary arteries after consumption of fish oils [23]. The

favorable effects of DHA on reducing arterial stiffness are also observed through direct vasodilation and inhibition of vasoconstrictor response in human studies [24]. In addition to these favorable structural effects of ω−3 fatty acids on the vascular system, there are some favorable functions as well. Improved endothelial-dependent dilation is one of the important vessel wall functions of ω−3 fatty acids which is achieved by increasing NO production [25] and decreasing oxidative stress [26].

It has been observed that ω−3 fatty acids are independently associated with a lower level of pro-inflammatory markers [IL-6, IL-1ra, tumor necrosis factor-a (TNFa) and CRP] and a higher level of anti-inflammatory markers [soluble IL-6r, IL-10, transforming growth factor-a (TGFa)] [27]. Hence, the presence of ω−3 fatty acids is beneficial in inflammatory-related diseases like atherosclerosis.

On the contrary, ω-6 fatty acids such as AA metabolites (prostaglandin 2, thromboxane 2, and leukotriene 4) promote prothrombotic and proinflammatory markers (IL-1, IL-6, NFKB, and TNF). Thus leading to inflammation, [28] thrombosis, atherosclerotic properties and contributing to the development of inflammatory disorders and excessive cell proliferation [29]. Therefore, levels of ω-6 fatty acids in vessels could be detrimental to health. In the present study, the percentage of the pro-inflammatory, arachidonic acid levels was significantly low in LIMA, compared to SV and RA suggesting a possible higher success rate in LIMA grafts.

The level of ω-3 fatty acid (Docosatetraenoic acid) observed in this study did not indicate a significant variation among the vessel types. However, in most of the studies, the ratio between ω-6 to ω-3 has been reported as an indication of the risk of atherosclerosis [12]. The observed ratio in the present study indicates a significantly low value for LIMA compared to the other two vessels. When the ratio of ω-6: ω -3 PUFA, increases it could potentiate inflammatory processes and consequently predispose to inflammatory diseases including atherosclerosis [20]. Therefore, the fatty acid profiles studied in LIMA, SV, and RA in the present study highlight the favorable fatty acid distribution of LIMA compared to RA and SV for CABG.

## Limitations

The number of specimens collected varies with the surgical procedure; hence in some patients, all three different samples could not be collected.

## Conclusions

There are significant differences observed in the percentage of fatty acids present in LIMA, SV, and RA biopsy samples obtained from CABG patients. Our results suggest that the risk of the redevelopment of atherosclerosis is less with LIMA when compared to SV and RA based on fatty acid composition.

## Supporting information

**S1 Table. Percentages fatty acids in LIMA.**
(DOCX)

**S2 Table. Percentages fatty acids in SV.**
(DOCX)

**S3 Table. Percentages fatty acids in RA.**
(DOCX)

**S4 Table. Linear regression analysis.**
(DOCX)

## Acknowledgments

Instrument center, Faculty of Applied Sciences, Department of Medical Laboratory Sciences, Faculty of Allied Health Sciences, University of Sri Jayewardenepura for providing the laboratory facilities and Nawaloka Hospital (PLC) is acknowledge for providing study samples.

## Author Contributions

**Conceptualization:** E. M. S. Bandara, D. D. C. de S. Wanniarachchi, H. Peiris, P. P. R. Perera, L. G. Chandrasena.

**Data curation:** D. I. U. Edirisinghe.

**Formal analysis:** E. M. S. Bandara, D. I. U. Edirisinghe, D. D. C. de S. Wanniarachchi, H. Peiris, L. G. Chandrasena.

**Funding acquisition:** D. D. C. de S. Wanniarachchi, L. G. Chandrasena.

**Investigation:** D. I. U. Edirisinghe, D. D. C. de S. Wanniarachchi, A. G. Jayakrishan, H. D. Waikar, S. K. Sharma, V. Abeysuriya, L. G. Chandrasena.

**Methodology:** E. M. S. Bandara, D. D. C. de S. Wanniarachchi, H. Peiris, P. P. R. Perera, L. G. Chandrasena.

**Project administration:** D. D. C. de S. Wanniarachchi, H. Peiris.

**Resources:** P. P. R. Perera, A. G. Jayakrishan, H. D. Waikar, S. K. Sharma, V. Abeysuriya, L. G. Chandrasena.

**Supervision:** E. M. S. Bandara, L. G. Chandrasena.

**Visualization:** D. I. U. Edirisinghe.

**Writing – original draft:** E. M. S. Bandara, D. I. U. Edirisinghe, D. D. C. de S. Wanniarachchi.

**Writing – review & editing:** D. D. C. de S. Wanniarachchi, H. Peiris, P. P. R. Perera, A. G. Jayakrishan, H. D. Waikar, S. K. Sharma, V. Abeysuriya, L. G. Chandrasena.

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
