## [Decision Letter · Decision Letter 0]

29 Jul 2021

PONE-D-21-17715

A comparative study on fatty acid profile in selected vessels of coronary artery bypass graft (CABG)

PLOS ONE

Dear Dr. Wanniarachchi,

Thank you for submitting your manuscript to PLOS ONE. After careful consideration, we feel that it has merit but does not fully meet PLOS ONE’s publication criteria as it currently stands. Therefore, we invite you to submit a revised version of the manuscript that addresses the points raised during the review process.

All reviewers and editors found that this manuscript is interesting and addresses a significant clinical problem, but the reviewers identified some important issues. These comments include small sample number, methodological problems, and English writing. The editors concur. Suggestions are all good ones and very constructive. Most of them can be addressed without additional analyses. The editor suggests that the authors should discuss some of the issues as limitations and de-emphasize the authors’ conclusion. If you are able to respond adequately to the points by the reviewers and editors, we are willing to reassess a revised manuscript; however, we cannot commit ourselves at this time to its publication.

We look forward to receiving your revised manuscript.

Kind regards,

Michinari Nakamura, MD

Academic Editor

PLOS ONE

Journal Requirements:

The work was supported by an institutional research grant from the University of Sri Jayewardenepura (Grant number ASP/01/RE/SCI/2019/67) awarded to D.D.C. de S.Wanniarachchi and funding received to E.M.S.Bandara from Nawaloka Hospital (PLC) Sri Lanka

Reviewers' comments:

Reviewer's Responses to Questions

**Comments to the Author**

1. Is the manuscript technically sound, and do the data support the conclusions?

Reviewer #1: Yes

Reviewer #2: Yes

Reviewer #3: No

2. Has the statistical analysis been performed appropriately and rigorously? 

Reviewer #1: Yes

Reviewer #2: Yes

Reviewer #3: No

3. Have the authors made all data underlying the findings in their manuscript fully available?

Reviewer #1: Yes

Reviewer #2: Yes

Reviewer #3: No

4. Is the manuscript presented in an intelligible fashion and written in standard English?

Reviewer #1: Yes

Reviewer #2: Yes

Reviewer #3: No

5. Review Comments to the Author

Reviewer #1: This manuscript describes that the analyses of fatty acid profile composition in major arteries which are commonly used in Coronary Artery Bypass Graft surgery (CABG) by GC-MS approach. This work provides valuable clinical information about the fatty acid profile composition on saphenous vein, radial artery and left internal mammary artery from patients undergoing elective CABG.

My comments are as follows:

1. More effort needs to be put on English editing.

2. How the percentage of each individual fatty acid was calculated? More detail descriptions are needed. How the samples are normalized?

3. Are the size numbers for all LIMA, SV and RA samples 27?

Reviewer #2: The present study reports comparison of fatty acid composition of three biopsy specimens of blood

vessels collected during CABG that include LIMA, RA, and SV vessels. The studies are well conducted albeit limited samples . They report the presence of free cholesterol and cholesterol esters in human atherosclerotic plaques . The findings suggest that ( based on fatty acid composition ) RA samples have more atherogenic activity compared to the SV and LIMA. These results may suggest that the risk of redevelopment of atherosclerosis is less with LIMA when compared to SV and RA (based on the fatty acid composition) for future CABG treatments.

The study is interesting and highly relevant to CABG treatment despite some limitations. The study is sound but it is unclear if the limited number of patient samples can allow them to reach definite conclusions .

Some grammatical errors do appear in the manuscript which needs attention.

Reviewer #3: In the study titled, "A comparative study on fatty acid profile in selected vessels of coronary artery bypass graft (CABG)," the investigators set out to compare the fatty acid composition of saphenous veins, radial arteries, and left internal mammary arteries that are being prepared as grafts for CABG. There are a number of major concerns with the structure of this manuscript in its current format. Logic is extremely hard to follow throughout.

1) The authors draw a conclusion that fatty acid composition of the LIMA makes it the best graft used for CABG, yet they provide no data on the relationship between fatty acid composition of a vessel and its ability to maintain graft patency. Nor do they provide any mechanism to support why fatty acid composition would influence graft patency.

2) Introduction: 3rd sentence structure is confusing as written.

3) Introduction: The logic is difficult to follow throughout the introduction. For example, the second paragraph attempts to introduce the concept that fatty acid composition is important to vessels, but concludes with a non sequitur about endothelial dysfunction as an indicator of atherosclerosis.

4) Introduction: Several component statements lack appropriate ciations.

a. “Fatty acid composition of vessel walls is an important modulator in vascular function.”

b. “Changes in fatty acids composition can modify the properties of cell membranes including fluidity and permeability, either directly via their insertion in the membrane or indirectly through their effects on lipid metabolism.”

5) Introduction: The third paragraph is unclear as written. Are the authors trying to say that saturated fats are good or bad? Are the authors trying to say thay poly-unsaturated fats are good or bad? There is no clear logic path in the way this paragraph is written.

6) Introduction: “Even though the fatty acid profile of SV, and LIMA had been studied before, there was no information on comparison of these profiles with radial artery.” Citations?

7) Introduction: Overall, does not clearly lay out the knowledge gap or what the goals of this study are.

8) Methodology: The statistical approach is not clearly written. Exactly what statistical methods were used to employ comparison between the three groups (LIMA, RA, and SV)? How were results adjusted for confounding variables?

9) Methodology: Numbers were small. How was this study powered?

10) Results: Table 1 is insufficient in terms of the data required for demographics and clinical characteristics. Need more than just Age, Sex, Hypertension, and Diabetes. Need information on: Age, BMI, Sex, DM, Hypertension, Hyperlipidemia, Cholesterol levels, Statin use, Smoking history, Family history, CAD, prior MI, GFR, ASCVD risk score, CVA.

11) Results: Table 2: what statistical test(s) were used to compare the 3 group and derive the P values?

12) Results: Table 3: what statistical test(s) were used to compare the 3 group and derive the P values?

13) The conclusion: “The fatty acid composition of RA indicates more atherogenic activity compared to the SV and LIMA. Our results suggest that the risk of redevelopment of atherosclerosis is less with LIMA when compared to SV and RA based on the fatty acid composition.” - The authors do not present any data to support this conclusion. This is not a study of graft patency or mechanisms of graft patency.

14) The study discussion indicates that the investigators assume the fatty acids in the vessels all come from atherosclerotic plaques within the vessel walls, yet there is no data to support these vessels, being used for bypass, had atherosclerosis.

15) There is no structure to the discussion; the rationale is difficult to follow.

16) The discussion section is missing a careful assessment of the study limitations.

17) Frequent English grammatical errors throughout.

6. PLOS authors have the option to publish the peer review history of their article (what does this mean?). If published, this will include your full peer review and any attached files.

Reviewer #1: No

Reviewer #2: **Yes: **Muthu Periasamy

Reviewer #3: No

---

## [Author Response · Author response to Decision Letter 0]

11 Sep 2021

PlosOne - Reviewer comments

Reviewer Comment Revision

General comments 1. Please ensure that your manuscript meets PLOS ONE's style requirements, including those for file naming. The PLOS ONE style templates can be found at 

Check ed.

The work was supported by an institutional research grant from the University of Sri Jayewardenepura (Grant number ASP/01/RE/SCI/2019/67) awarded to D.D.C. de S.Wanniarachchi and funding received to E.M.S.Bandara from Nawaloka Hospital (PLC) Sri Lanka.

 Funding statement is amended to include The funders had no role in study design, data collection and analysis, decision to publish, or preparation of the manuscript

 3. Your ethics statement should only appear in the Methods section of your manuscript. If your ethics statement is written in any section besides the Methods, please move it to the Methods section and delete it from any other section. Please ensure that your ethics statement is included in your manuscript, as the ethics statement entered into the online submission form will not be published alongside your manuscript. Ethics statement is only written under the methods section

Reviewer 1 This manuscript describes that the analyses of fatty acid profile composition in major arteries which are commonly used in Coronary Artery Bypass Graft surgery (CABG) by GC-MS approach. This work provides valuable clinical information about the fatty acid profile composition on saphenous vein, radial artery and left internal mammary artery from patients undergoing elective CABG.

Comment 1 More effort needs to be put on English editing Grammar checked and revised. 

Comment 2 How the percentage of each individual fatty acid was calculated? More detail descriptions are needed. How the samples are normalized? The percentage of individual fatty acids was calculated as the area percentage of each fatty acid to the total area of fatty acids detected. 

Comment 3 Are the size numbers for all LIMA, SV and RA samples 27? Grafting vessels selected from patients vary depending on the surgical procedure. Hence two or three different vessel samples were obtained from a patient. Therefore, among 27 patients selected in the study LIMA = 12, SV = 20 RA = 9 samples were analyzed.

Reviewer 2 The present study reports comparison of fatty acid composition of three biopsy specimens of blood

vessels collected during CABG that include LIMA, RA, and SV vessels. The studies are well conducted albeit limited samples . They report the presence of free cholesterol and cholesterol esters in human atherosclerotic plaques . The findings suggest that ( based on fatty acid composition ) RA samples have more atherogenic activity compared to the SV and LIMA. These results may suggest that the risk of redevelopment of atherosclerosis is less with LIMA when compared to SV and RA (based on the fatty acid composition) for future CABG treatments.

.

Comment 1 The study is interesting and highly relevant to CABG treatment despite some limitations. The study is sound but it is unclear if the limited number of patient samples can allow them to reach definite conclusions The sample size was estimated with single group mean (N = (Zα/2)2 s2 / d2) .

Considering the standard deviation of Docosahexaenoic acid level of internal mammary artery (reference number 8). 

Hence, Zα/2 = 1.96, s= 2.5, d=1

N=24

The corrected sample size after 10% allowance of missing is 27.

Comment 2 Some grammatical errors do appear in the manuscript which needs attention. Grammar revised. and corrections are highlighted

Reviewer 3 In the study titled, "A comparative study on fatty acid profile in selected vessels of coronary artery bypass graft (CABG)," the investigators set out to compare the fatty acid composition of saphenous veins, radial arteries, and left internal mammary arteries that are being prepared as grafts for CABG. There are a number of major concerns with the structure of this manuscript in its current format. Logic is extremely hard to follow throughout.

Comment 1 The authors draw a conclusion that fatty acid composition of the LIMA makes it the best graft used for CABG, yet they provide no data on the relationship between fatty acid composition of a vessel and its ability to maintain graft patency. Nor do they provide any mechanism to support why fatty acid composition would influence graft patency. The graft patency depends on multiple parameters where fatty acid of composition of grafting blood vessels is also an important parameter. Therefore the study is focused on analysis fatty acid composition since there are limited studies conducted. Support by literature (Petruzzo

et al., 2001). Included in the discussion. 

Comment 2 Introduction: 3rd sentence structure is confusing as written. Corrected as “The majority of patients receive left internal mammary artery (LIMA) grafts to the left anterior descending (LAD) coronary artery compared to other grafting options such as saphenous vein (SV) grafts or radial artery (RA)”.

Comment 3 Introduction: The logic is difficult to follow throughout the introduction. For example, the second paragraph attempts to introduce the concept that fatty acid composition is important to vessels, but concludes with a non sequitur about endothelial dysfunction as an indicator of atherosclerosis. Corrected as “Hence, several studies has observed the influence of the vessel wall fatty acid composition in atherosclerosis”.

Comment 4 Introduction: Several component statements lack appropriate ciations.

a. “Fatty acid composition of vessel walls is an important modulator in vascular function.”

b. “Changes in fatty acids composition can modify the properties of cell membranes including fluidity and permeability, either directly via their insertion in the membrane or indirectly through their effects on lipid metabolism.” References are included in the paper.

Comment 5 Introduction: The third paragraph is unclear as written. Are the authors trying to say that saturated fats are good or bad? Are the authors trying to say thay poly-unsaturated fats are good or bad? There is no clear logic path in the way this paragraph is written. The third paragraph is revised to emphasis the role of fatty acids in cell walls in developing atherosclerosis.

Comment 6 Introduction: “Even though the fatty acid profile of SV, and LIMA had been studied before, there was no information on comparison of these profiles with radial artery.” Citations? References included.

Comment 7 Introduction: Overall, does not clearly lay out the knowledge gap or what the goals of this study are. The introduction section is revised to emphasize the knowledge gap and highlighting the goal of the study. 

Comment 8 Methodology: The statistical approach is not clearly written. Exactly what statistical methods were used to employ comparison between the three groups (LIMA, RA, and SV)? How were results adjusted for confounding variables? The data was analyzed using SPSS version 25. Since the Kolmogorov–Smirnov test reported the variables are not normally distributed, Mann–Whitney U-tests were used to compare the fatty acid percentage of LIMA, SV and RA. A p value of <0.05 was considered to be significant. 

A linear regression models were developed to determine the association of confounding variable with fatty acid level. The results included in the manuscript and detail analysis were included in the supporting information. 

Comment 9 Methodology: Numbers were small. How was this study powered? The sample size was estimated with single group mean (N = (Zα/2)2 s2 / d2) .

Considering the standard deviation of Docosahexaenoic acid level of internal mammary artery (reference number 8). 

Hence, Zα/2 = 1.96, s= 2.5, d=1

N=24

The corrected sample size after 10% allowance of missing is 27. 

Comment 10 Results: Table 1 is insufficient in terms of the data required for demographics and clinical characteristics. Need more than just Age, Sex, Hypertension, and Diabetes. Need information on: Age, BMI, Sex, DM, Hypertension, Hyperlipidemia, Cholesterol levels, Statin use, Smoking history, Family history, CAD, prior MI, GFR, ASCVD risk score, CVA. Table 1 is revised to include further information suggested.

Comment 11 Results: Table 2: what statistical test(s) were used to compare the 3 group and derive the P values? Mann–Whitney U-tests were used to compare the fatty acid percentage of LIMA, SV and RA. A p value of <0.05 was considered to be significant.

Comment 12 Results: Table 3: what statistical test(s) were used to compare the 3 group and derive the P values? Mann–Whitney U-tests were used to compare the fatty acid percentage of LIMA, SV and RA. A p value of <0.05 was considered to be significant.

Comment 13 The conclusion: “The fatty acid composition of RA indicates more atherogenic activity compared to the SV and LIMA. Our results suggest that the risk of redevelopment of atherosclerosis is less with LIMA when compared to SV and RA based on the fatty acid composition.” - The authors do not present any data to support this conclusion. This is not a study of graft patency or mechanisms of graft patency. The main focus of the paper is to find the presence of fatty acids which would attribute to the development of plaque. It is not a study of graft patency. The conclusions were drawn from the perspective of fatty acid composition. 

Comment 14 The study discussion indicates that the investigators assume the fatty acids in the vessels all come from atherosclerotic plaques within the vessel walls, yet there is no data to support these vessels, being used for bypass, had atherosclerosis. The study samples of SV,RA,and LIMA are left over parts of vessels used in CABG during surgery. These do not contained plaques.

Comment 15 There is no structure to the discussion; the rationale is difficult to follow. Discussion section was revised

Comment 16 The discussion section is missing a careful assessment of the study limitations. Study limitations were mentioned.

Comment 17 Frequent English grammatical errors throughout. Grammar revised.

---

## [Decision Letter · Decision Letter 1]

17 Nov 2021

A comparative study on fatty acid profile in selected vessels of coronary artery bypass graft (CABG)

PONE-D-21-17715R1

Dear Dr. Wanniarachchi,

We’re pleased to inform you that your manuscript has been judged scientifically suitable for publication and will be formally accepted for publication once it meets all outstanding technical requirements.

Kind regards,

Michinari Nakamura, MD

Academic Editor

PLOS ONE

Additional Editor Comments (optional):

Reviewers' comments:

Reviewer's Responses to Questions

**Comments to the Author**

1. If the authors have adequately addressed your comments raised in a previous round of review and you feel that this manuscript is now acceptable for publication, you may indicate that here to bypass the “Comments to the Author” section, enter your conflict of interest statement in the “Confidential to Editor” section, and submit your "Accept" recommendation.

Reviewer #1: All comments have been addressed

2. Is the manuscript technically sound, and do the data support the conclusions?

Reviewer #1: Yes

3. Has the statistical analysis been performed appropriately and rigorously? 

Reviewer #1: Yes

4. Have the authors made all data underlying the findings in their manuscript fully available?

Reviewer #1: Yes

5. Is the manuscript presented in an intelligible fashion and written in standard English?

Reviewer #1: No

6. Review Comments to the Author

Reviewer #1: Majority questions were addressed and the work also provides valuable clinical information. However, the quality of English is not good enough for publication.

7. PLOS authors have the option to publish the peer review history of their article (what does this mean?). If published, this will include your full peer review and any attached files.

Reviewer #1: No

---

## [Editor Report · Acceptance letter]

1 Dec 2021

PONE-D-21-17715R1 

A comparative study on fatty acid profile in selected vessels of coronary artery bypass graft (CABG) 

Dear Dr. Wanniarachchi:

I'm pleased to inform you that your manuscript has been deemed suitable for publication in PLOS ONE. Congratulations! Your manuscript is now with our production department. 

Kind regards, 

on behalf of

Dr. Michinari Nakamura 

Academic Editor

PLOS ONE